# Growth and Overall Health of Patients with *SLC13A5* Citrate Transporter Disorder

**DOI:** 10.3390/metabo11110746

**Published:** 2021-10-29

**Authors:** Tanya L. Brown, Kimberly L. Nye, Brenda E. Porter

**Affiliations:** 1Treatments for Epilepsy and Symptoms of SLC13A5 Foundation, TESS Research Foundation, Menlo Park, CA 94026, USA; kim@tessfoundation.org; 2Department of Neurology and Neurological Sciences, Stanford University, Palo Alto, CA 94070, USA; brenda2@stanford.edu

**Keywords:** *SLC13A5*, citrate, NaCT, transporter

## Abstract

We were interested in elucidating the non-neurologic health of patients with autosomal recessive *SLC13A5* Citrate Transporter (NaCT) Disorder. Multiple variants have been reported that cause a loss of transporter activity, resulting in significant neurologic impairment, including seizures, as well as motor and cognitive dysfunction. Additionally, most patients lack tooth enamel (amelogenesis imperfecta). However, patients have not had their overall health and growth described in detail. Here we characterized the non-neurologic health of 15 patients with medical records uploaded to Ciitizen, a cloud-based patient medical records portal. Ciitizen used a query method for data extraction. Overall, the patients’ records suggested a moderate number of gastrointestinal issues related to feeding, reflux, vomiting and weight gain and a diverse number of respiratory complaints. Other organ systems had single or no abnormal diagnoses, including liver, renal and cardiac. Growth parameters were mostly in the normal range during early life, with a trend toward slower growth in the few adolescent patients with data available. The gastrointestinal and pulmonary issues may at least partially be explained by the severity of the neurologic disorder. More data are needed to clarify if growth is impacted during adolescence and if adult patients develop or are protected from non-neurologic disorders.

## 1. Introduction

Over 40 different variants (https://www.tessresearch.org/research-tools/#pathogenic-mutations, accessed on 27 October 2021), have been reported in the *SLC13A5* gene. All 40 variants are associated with an autosomal recessive disorder causing a distinct neurologic phenotype; multiple seizure semiologies are reported, with epilepsy starting during the neonatal period, along with significant abnormalities in motor and cognitive function. While not all described variants have undergone citrate transport measurements, loss of citrate transport has been described in all the variants tested to date [1,2,3]. The *SLC13A5* gene and its transcribed protein, NaCT, are highly expressed in the liver at levels thought to be several-fold higher than in the brain [4]. However, the loss of citrate transport, presumably throughout the body, has only been associated with neurologic and dental abnormalities in humans [5,6,7,8,9]. Citrate levels were elevated in the CSF and plasma of the few patients tested [10]. It is unclear whether the elevated extracellular citrate levels contribute to the neurologic phenotype or cause other non-neurologic sequelae.

While the neurologic phenotype has been described in a mouse model with loss of *SLC13A5*, a variety of animal models also show robust non-neurologic phenotypes. Reduced expression of *SLC13A5*, **I** am **N**ot **D**ead **Y**et (INDY), in flies and worms increases lifespan and decreases whole-body fat [11,12,13]. Mice lacking *SLC13A5* show increased propensity for seizures as part of the neurologic phenotype [14]. In addition to a neurologic phenotype, loss of *SLC13A5* in mice reduces growth parameters such as weight and length [15,16]. Loss of *SLC13A5* in mice also reduces blood pressure and heart rate [16]. These data suggest that reduced citrate transport may have widespread consequences beyond the previously described neurologic phenotype in the patients.

Furthermore, the importance of citrate transport deficiency extends beyond patients with *SLC13A5* Citrate Transporter Disorder, due to interest in developing liver specific NaCT inhibitors to treat metabolic syndrome and extending the lifespan [14,17,18,19,20,21,22]. The overall health of patients with *SLC13A5* Citrate Transporter Disorder might elucidate sequela of NaCT inhibition therapy. Increased understanding of the non-neurologic pathophysiology of citrate transport loss is needed to improve understanding of the role of citrate in health.

We analyzed the medical records from 15 patients with clinical and genetic diagnosis of *SLC13A5* Citrate Transporter Disorder, identified by their neurologic phenotype, clinical genetic testing and genomic analysis. All patients’ families participated by allowing Ciitizen to access and collect the medical records of the patients. The records were then queried by Ciitizen using automated system capture to extract specific information and deidentified data were provided to the TESS Research Foundation for further analysis. Our findings demonstrated normal growth for *SLC13A5* Citrate Transporter Disorder patients in the first three years of life and that patients had mild health complications outside the nervous system.

## 2. Results

### 2.1. SLC13A5 Citrate Transporter Disorder Patient Demographics

*SLC13A5* Citrate Transporter Disorder patients included in this study include both male and female patients ranging from 1 to 17 years of age from 11 different families (Table 1). There is a unique range of medical records collected for each patient. Overall, there are 16 distinct DNA variants, most frequently with a single nucleotide change. There are 12 unique protein variants, 1 complete protein deletion and 3 variants with unknown effects on protein. The three variants with unknown protein consequences include two splice site variants and one partial gene deletion (Table 1). Patients born before 2014, when *SLC13A5* Citrate Transporter Disorder was first identified, often lacked a conclusive diagnosis for the etiology of their neonatal seizures and developmental delay. For patients born before 2014, the average age of diagnosis was at 6.1 years (standard deviation +/− 1.96 years). However, after this disorder was identified, most patients underwent genetic testing before 1 year of age. Earlier diagnosis may provide increased opportunity for effective treatment, particularly with the advent of precision therapy.

### 2.2. Growth Parameters

#### 2.2.1. *SLC13A5* Citrate Transporter Disorder Patient Physical Development 0–36 Months

Citrate is a key metabolite and an important factor in regulating energy expenditure. *SLC13A5* Citrate Transporter Disorder patients have elevated plasma and CSF citrate levels, indicating an inability to transport citrate into cells. *SLC13A5* citrate transporter patients also show altered metabolic profiles [10]. Since metabolism is important for cellular function and growth, we assessed growth parameters during early development. As patients begin experiencing seizures within the first days after birth, we first assessed growth (head circumference, weight, height) from birth to 36 months. We used the Centers for Disease Control and Prevention National Center for Health Statistics clinical growth charts (CDC growth charts) to assess how *SLC13A5* Citrate Transporter Disorder patients developed over time. We found that both male and female patients showed grossly normal head growth (Figure 1A,B). We then assessed weight and height during the same period. As with head circumference results, we found that *SLC13A5* Citrate Transporter Disorder patients showed grossly normal weight and height measurements for age (Figure 2). Despite decreased citrate transport and altered metabolic profiles, the patients’ growth, weight and height were within normal parameters during the first three years of life.

#### 2.2.2. *SLC13A5* Citrate Transporter Disorder Patient Physical Development 2–20 Years

As *SLC13A5* Citrate Transporter Disorder patients age, they experience increasingly severe motor and cognitive delays, along with ongoing seizures [6,7]. Patient weight and height from 2–20 years of age were compared with CDC normative data. Additionally, we calculated the body mass index (BMI). We observed both male and female patients until 10 years of age, mostly growing within the 5th to 95th percentiles (Figure 3A–D). We have limited data from patients older than 7 years of age. However, in the older patient data shown here, both males and females (Figure 3C,D) crossed several isopleths after 5 years of age, with two older patients eventually falling below the 5th percentile.

### 2.3. Non-Neurologic Diagnoses

Clinical diagnoses were culled by one of the authors, BEP, and those related to disorders outside of the nervous system are listed in Table 2, with the patients having the diagnosis listed. A total of 309 distinct diagnoses were reported at least once in the records. There was a wide range of diagnoses reported per patient. The fewest was five diagnoses for Female 7 and a maximum of 41 diagnoses reported for Male 3, with a mean of 20.4 diagnoses per patient. Gastroesophageal reflux was the most common diagnosis in five patients, closely followed by feeding difficulties in four patients. Apnea was reported in four patients. Five patients had diagnoses related to growth or feeding issues, which correlate with delayed growth parameters of the patients Female 2, 3 and 6 and Male 7, as shown in Figure 2.

### 2.4. Diagnostic Procedures

Patients underwent many diagnostic procedures (299 total), but most were directly related to the evaluation of the nervous system, including EEGs (171), and brain imaging studies (62). Thirty-one procedures were performed to evaluate non-neurologic organ systems and are listed in Table 3. Abnormal studies of the gastrointestinal system, pulmonary and cardiac system were reported in a small subset of the patients.

### 2.5. Surgical Procedures

Patients underwent a small number of surgical procedures unrelated to the nervous system, including three patients with placement of a gastrostomy tube, see Table 4.

## 3. Discussion

We provided evidence of mostly mild health conditions outside of the nervous system, in 15 pediatric patients from the US with *SLC13A5* Citrate Transporter Disorder. Deaths and severe structural or non-neurologic diseases were absent from the medical records of the patients. 

There was no diagnosis that occurred in most of the patients, though several had a mixture of gastrointestinal and pulmonary diagnoses. There were rare reports for cardiovascular, hepatobiliary, dental and urinary disorders, both by diagnoses and diagnostic testing. There was concordance with patients having both gastrointestinal diagnoses, including feeding difficulties, gastroesophageal reflux, abnormal swallowing studies and placement of gastrostomy tubes. Although dental records were not collected, several patients included dental diagnosis abnormalities. It will be important for future studies to determine whether the dental abnormalities and gastrointestinal diagnoses contribute to a decrease in growth, independent of the lack of citrate transport. 

Additionally, patients had respiratory diagnoses, including apnea, snoring, abnormal sleep studies and patients underwent tonsillectomies and adenoidectomies. There is an overlap in the gastrointestinal and pulmonary diagnoses from our patients’ medical records and those often associated with other neurologic disorders, such as cerebral palsy and other rare epilepsy syndromes, suggesting that neurologic dysfunction may be a strong contributing factor [23,24,25].

There was a lack of diagnoses associated with laboratory abnormalities, such as renal or liver disorders, which indirectly suggest that those organs are overall functioning normally in this population. However, we could not confirm this due to gaps in the data extracted by Ciitizen, including labs and vital signs that were not extracted. To determine whether patients are protected from obesity and metabolic syndrome, a much larger data pool with older patients is required.

Growth parameters such as height, weight and head growth appear normal at birth and throughout early life for most patients. The literature suggests microcephaly is rare, at less than 10% of patients. Our findings of normal head growth early in life are consistent with mostly intact head growth [1,6]. One patient, Female 3, had a diagnosis of prematurity, but others lacked this diagnosis and had growth parameters suggesting a full-term infant. Patients with extensive measurements through late childhood and adolescence began to cross isopleths and eventually, while still growing, two were below the 5th percentile. Further studies of growth parameters will be important to understand whether this is a common finding and whether the etiology is central to the disorder or multifactorial, due to the feeding issues surrounding their dental and neurologic health.

The data presented here show a similar trend to animal and in vitro models, with a loss of *SLC13A5* resulting in altered growth. In mice, loss of *SLC13A5* leads to lower body weight compared to controls after 4 weeks of age [14,15]. Intriguingly, body weight differences continued to expand up to 36 weeks of age [15]. *SLC13A5*−/− mice also show a slight but conceptually significant decreased body length at 3 months of age [15]. The decrease in growth aligns with in vitro models demonstrating that, after knockdown of *SLC13A5* in HepG2 cells, there is upregulation of ketogenesis, fatty acid β oxidation and catabolic process, ultimately limiting lipid accumulation [22]. Altered INDY function in drosophila and c elegans causes an improvement in longevity but reported homozygous mutations are not as helpful to overall health, suggesting that there is a dosage effect [26]. As more patients are identified, particularly older patients born prior to the availability of molecular diagnosis, it will be important to assess growth during adolescence.

In addition to growth inhibition, rodents with loss of *SLC13A5* had decreased blood pressure and heart rates [16]. It will be important for future studies to investigate whether *SLC13A5* Citrate Transporter Disorder patients show differences in cardiovascular function. Detailed characterization of the non-neurologic presentation of *SLC13A5* Citrate Transporter Disorder may have implications for therapeutic targets and biomarker development of future treatments.

The use of medical records for case series has a long history in biomedical research. We utilized a novel approach of data capture using the medical record system transmitted to a third party for data abstraction. The families of the patients provided access to the medical records for collection by an outside company, Ciitizen. The records were then extracted using key words and provided as a spreadsheet. This introduced multiple biases due to an inability to ensure complete data collection; of note, several patients had missing data at different ages. Nonsystematic comparisons of the machine-extracted results to the raw clinical data in limited cases confirmed the accuracy of medications and growth parameters.

## 4. Materials and Methods

### 4.1. Medical Record Collection and Data Extraction

Caregivers of children and adults with a confirmed diagnosis of *SLC13A5* Citrate Transporter Disorder were invited to join the Ciitizen/TESS Research Foundation Databank, which received an IRB determination of exemption. Ciitizen is a patient-facing platform that collects designated record sets by leveraging the HIPPA right of access. Record completion was verified through evaluation of document class and encounter year. Records included medical records. Dental records were not collected.

Ciitizen has developed a proprietary approach to streamline the generation of regulatory-grade clinical data from unstructured data sources. In brief, medical records are ingested into the platform for document preprocessing. Through a series of artificial intelligence services, document attributes are determined and subsequently verified by human clinicians. To support systematic data capture and harmonization of data sources, Ciitizen extracts information longitudinally from each document in accordance with a data model, encompassing: genotype, clinical phenotype and therapeutic interventions, among others. Ciitizen has developed a curated ontology that supports mapping of extracted data to standard codes derived from internationally recognized terminologies. All extracted data are independently verified by two clinicians with relevant training. Ciitizen data, shared with TESS Research Foundation, exclude all personally identifying information about the family or its members.

### 4.2. Medical Record Analysis: Patient Growth

De-identified data from Ciitizen were analyzed using RStudio version 1.4.1106. Code used to analyze data can be found on GitHub (https://github.com/tanyab37/patient_growth, accessed on 27 October 2021). De-identified data points greater than 3 standard deviations or data with incorrect units were excluded from analysis. BMI was calculated using patient weight and height collected on the same date. Data files from the Centers for Disease Control and Prevention National Center for Health Statistics were used to plot percentile curves (https://www.cdc.gov/growthcharts/percentile_data_files.htm, accessed on 27 October 2021).

### 4.3. Medical Record Analysis: Patient Diagnoses

De-identified data from Ciitizen were manually evaluated. All non-neurologic diagnoses and procedures performed were incorporated into the included tables.

## 5. Conclusions

The presented data describe the non-neurologic health of patients with *SLC13A5* Citrate Transporter Disorder. Patients demonstrate a myriad of mild health complications outside the nervous system. These data demonstrate the power of using automated data extraction methods to garner information from medical records, in combination with manual analysis, to assess patient health. Further studies will need to assess growth during adolescence and beyond. 

## Figures and Tables

**Figure 1 metabolites-11-00746-f001:**
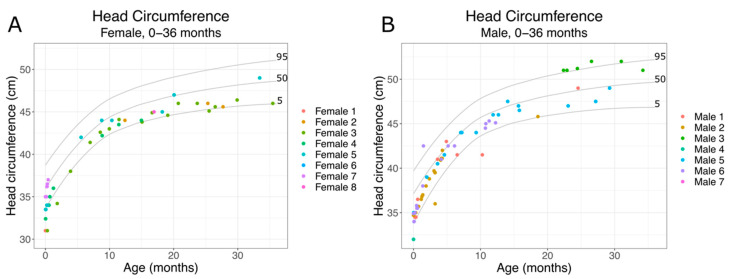
*SLC13A5* Citrate Transporter Disorder patient head circumference: head circumference from 0–36 months in female (**A**) and male (**B**) patients are shown with the 95th, 50th and 5th percentile head circumferences from the CDC growth charts.

**Figure 2 metabolites-11-00746-f002:**
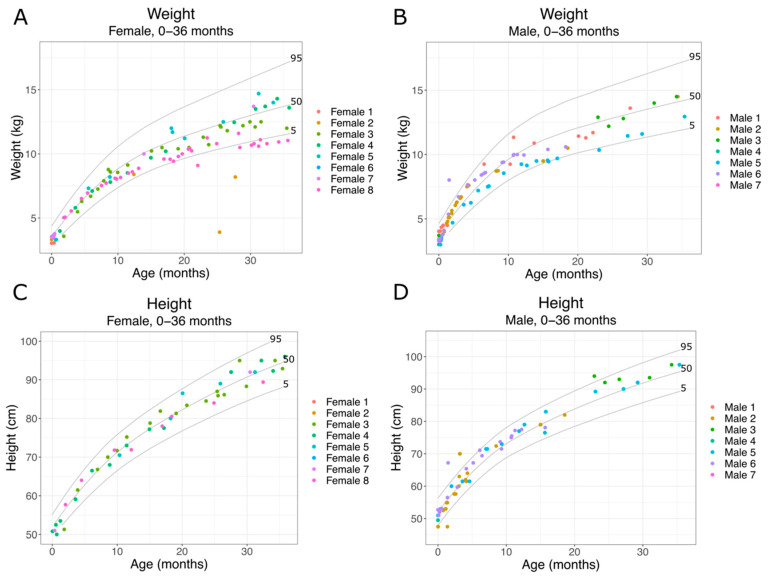
*SLC13A5* Citrate Transporter Disorder patient weight 0–36 months: (**A**) weight of female patients 0–36 months; (**B**) weight of male patients 0–36 months; (**C**) height of female patients 0–36 months; (**D**) height of male patients from 0–36 months. Each panel includes the 95th, 50th and 5th percentiles from the CDC growth charts.

**Figure 3 metabolites-11-00746-f003:**
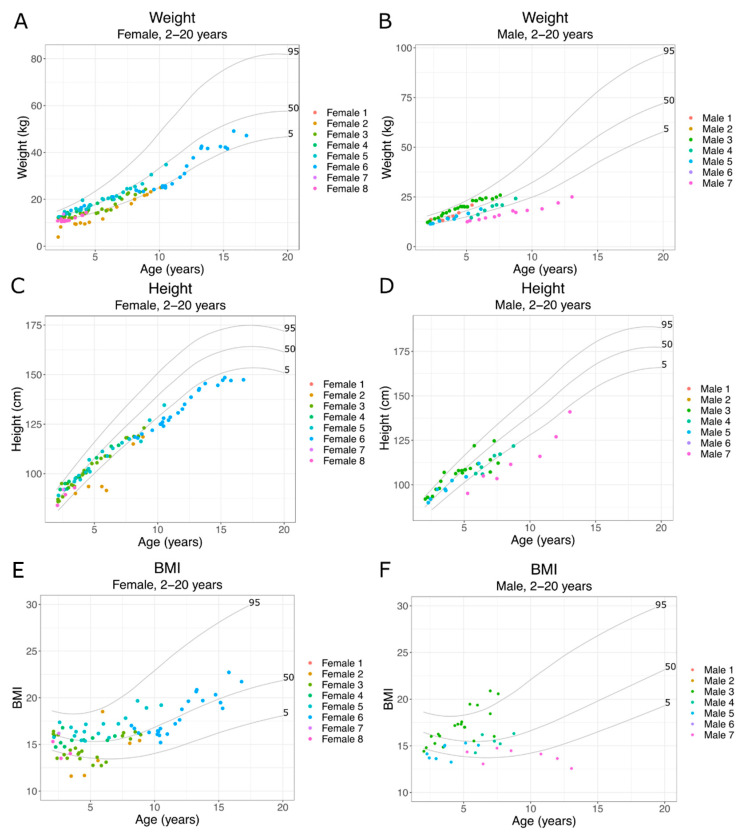
*SLC13A5* Citrate Transporter Disorder patient growth 2–20 years: (**A**) weight of female patients 2–20 years; (**B**) weight of male patients 2–20 years; (**C**) height of female patients 2–20 years old; (**D**) height of male patients from 2–20 years old; (**E**) BMI of female patients 2–20 years; (**F**) BMI of male patients 2–20 years. Each panel includes the 95th, 50th and 5th percentiles from the CDC growth charts.

**Table 1 metabolites-11-00746-t001:** *SLC13A5* Citrate Transporter Disorder patient demographics. ^a^ siblings, ^b^ siblings, ^c^ siblings, ^d^ siblings. Clinical significance based on ClinVar. P: pathogenic, LP: likely pathogenic, VUS: variant of uncertain significance, N: not found in ClinVar database at time of submission.

ID	DNA Variant	Protein Variant	Variant Classification	Clinical Significance (ClinVar)	Age at Diagnosis (Years)	Birth Year	Medical Records Collection
**Female 1**	c.1599C > A	p.Asn533Lys	missense	VUS	<1	2020	2020–2021
	c.650G > A	p.Ser217Asn	missense	VUS			
**Female 2 ^a^**	c.655G > A	p.Gly219Arg	missense	P/LP	4	2010	2010–2020
	c.245A > G	p.Tyr82Cys	missense	N			
**Female 3 ^b^**	c.511delG	p.Glu171Serfs*16	frameshift	P	7	2006	2006–2015
	c.511delG	p.Glu171Serfs*16	frameshift	P			
**Female 4 ^c^**	c.368 + 1G > A		splice site	N		2013	2013–2020
	c.368 + 1G > A		splice site	N			
**Female 5 ^c^**	c.368 + 1G > A		splice site	N	5	2008	2008–2020
	c.368 + 1G > A		splice site	N			
**Female 6 ^d^**	c.1475T > C	p.Leu492Pro	missense	LP	10	2003	2007–2020
	c.655G > A	p.Gly219Arg	missense	P/LP			
**Female 7**	Gene Deletion		gene deletion	N	<1	2018	2018–2021
	c.425C > T	p.Thr142Met	missense	LP			
**Female 8**	chr17:6606,915-6611,132 × 1		partial gene deletion	N	<1	2016	2016–2020
	c.389G > A	p.Gly130Asp	missense	N			
**Male 1**	c.997C > T	p.Arg333Ter	missense	P	6	2013	2013–2020
	c.997C > T	p.Arg333Ter	missense	P			
**Male 2**	c.232-2A > G		splice site	LP	<1	2019	2019–2020
	c.1460C > T	p.Pro487Leu	missense	VUS			
**Male 3**	c.1511delT	p.Leu504Cysfs*23	frameshift	N	4	2012	2012–2020
	c.1511delT	p.Leu504Cysfs*23	frameshift	N			
**Male 4 ^b^**	c.511delG	p.Glu171Serfs*16	frameshift	P	6	2007	2007–2016
	c.511delG	p.Glu171Serfs*16	frameshift	P			
**Male 5 ^d^**	c.655G > A	p.Gly219Arg	missense	P/LP	<1	2013	2013–2020
	c.1475T > C	p.Leu492Pro	missense	LP			
**Male 6**	c.1514C > T	p.Pro505Leu	missense	VUS	<1	2019	2019–2020
	c.997C > T	p.Arg333Ter	missense	P			
**Male 7 ^a^**	c.655G > A	p.Gly219Arg	missense	P/LP	7	2007	2012–2021
	c.245A > G	p.Tyr82Cys	missense	N			

**Table 2 metabolites-11-00746-t002:** All non-neurologic diagnoses are listed with each patient carrying that diagnosis on one or more occasion.

Gastrointestinal	Subjects
Feeding difficulty	F8; M1,3,6
Gastroesophageal reflux disease (GERD)	F1,5,8; M1,7
Abdominal pain	F2
Vomiting	M3,5,6
Projectile Vomiting	M2
Oropharyngeal dysphagia	F2; M1
Incontinence of feces	M3
**Urinary**	
Urinary incontinence	M3
Vesicoureteral reflux	F2
Recurrent urinary tract infections	F8; M4,6
Retention of urine	F8;
**Musculoskelatal**	
Talipes planovalgus	F3; M4
Trigonocephaly/Simple craniosynostosis	M1
Fracture of radius	F5
Calcaneovalgus deformity of foot	F3
Joint hypermobility	F3; M4
Varus deformity	F3
Simple craniosynostosis	M1
**Nutrition and growth**	
Slow weight gain	F2; M7
Short stature	F2,6; M7
Failure to thrive	F3
**Respiratory**	
Ineffective airway clearance	M3
Breathing-related sleep disorder	M1
Pulmonary aspiration	M1
Reactive airway disease	F2; M3
Respiratory distress	F2
Respiratory failure	M1
Respiratory insufficiency	F5
Restrictive lung disease	M1
Ineffective airway clearance	M3
Apnea	F3,5; M3,6
Acute bronchitis	M1
Acute respiratory failure	M6
Snoring	F8; M1,3
Chronic cough	M1
Atelectasis	M6
Vocal cord paralysis	F2
Chronic pulmonary aspiration	M3
**Cardiovascular**	
Patent ductus arteriosus	F3
Patent foramen ovale	F3
Mitral valve regurgitation	F2
Heart murmur	M1
**Hematologic**	
Anemia	F3
**Integument**	
Eczema	F8
**Dental Health**	
Impaired dentition	F3
Partial congenital absence of teeth	F2
Congenital anomaly in number of teeth	F5; M6

**Table 3 metabolites-11-00746-t003:** Non-neurologic studies performed and results. Each study performed is listed; if a patient is listed more than once that represents a separate procedure. Study findings are simplified and listed within the parentheses. GERD-gastrointestinal esophageal reflux disease.

Cardiac	Subjects
Echocardiography (Normal)	F3; M1,4
Echocardiography (Patent Foramen Ovale)	F3,5; M2
Electrocardiographic monitoring (Normal)	M1,4
**Bone health**	
Radiography for bone age studies (Normal)	F3; M4
**Gastrointestinal**	
Videofluoroscopic swallow study (Oropharyngeal dysphagia)	M1
Videofluoroscopic swallow study (Pulmonary aspiration)	M1
Videofluoroscopic swallow study (Laryngeal penetration)	M5
Ultrasonography of abdomen (Normal)	F3; M2,4
Upper gastrointestinal tract series (Normal)	M3
Upper gastrointestinal tract series (GERD)	F5
Diagnostic radiography of abdomen (Normal)	M2
**Renal**	
Ultrasonography of bilateral kidneys (Normal)	F8,F8,F8
Ultrasonography of bilateral kidneys (Bladder distention)	F8
Ultrasonography of bilateral kidneys (pyelectasia)	M6
Video urodynamic study (Normal)	F8
Voiding urethrocystography (Normal)	F3; M4
**Sleep**	
Polysomnogram (Obstructive sleep apnea of child)	F3; M1
Polysomnogram (Normal)	M1
Polysomnogram (abnormal)	F8

**Table 4 metabolites-11-00746-t004:** Procedures performed are listed and the patients receiving them are listed.

Procedure	Subjects
Gastrostomy tube placement	F2; M3,6
Myringotomy tube insertion	M3
Tonsil and adenoidectomy/adenoidectomy	F7; M3
Release of trigger thumb	F5
Incision of lingual frenum	M6

## Data Availability

Data used for this publication can be accessed by contacting research@ciitizen.com, accessed on 27 October 2021.

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
