# Peer review of "Growth and Overall Health of Patients with SLC13A5 Citrate Transporter Disorder"

_metabolites, 2021, doi:10.3390/metabo11110746_

Round 1

Reviewer 1 Report

In the manuscript "Growth and overall health of patients with SLC13A5 citrate transporter disorder", the authors provide an overall summary of the non-neurological health in the children with the respective disorder. As the neurological symptoms are extensively described in literature, the non-neurological health was under reported so far. As such, this manuscript is of very high significance to the field, is very well written and provides information on the individual patient level. However, a few questions remain unanswered.

  • In the introduction the authors mention the increase in lifespan when Indy is deleted in worms and flies. However, it should be noted that this was the case for heterozygotes as homozygote deletions rather cause deleterious effects
  • Rephrase sentence: "in mice, in addition to neurologic phenotype, loss of SLC13A5 ....". The added references do not report a neurological phenotype. 
  • Children with SLC13A5 disorder present with amelogenesis imperfecta but the authors did not elaborate on this aspect in the "non-neurological diagnosis" description. Does this affect the children's eating abilities? And to what extent is this "phenotype" similar to what has been described in mice? (Armando R. Irizarry et al., Plos One, 2017)
  • It is intriguing that SLC13A5 mutations consistently cause the neurological phenotypes. However, the gastrointestinal and respiratory symptoms are inconsistent. Are these direct effects from the SLC13A5 mutations on those organs and that in these few specific patients the compensation mechanisms fail too? Or are these secondary effects to treatments/diets/etc?
  • Is there a correlation between the type of mutation and the severity of the patient's health?

Reviewer 2 Report

In the paper titled "Growth and overall health of patients with SLC13A5 citrate transporter disorder"  the authors look at non-neurological data from patients with SLC13A5 citrate transport disorder.

Overall the manuscript is well written but needs minor editing for content. The github link provided by the authors does not work, please update the link.

It is also not clear from the methods section what kind of machine learning approaches were employed to extract patient data.

I would also like the authors to elaborate more on how the human SLC13A5 differs from other species specially flies and c. elegans where a deletion of INDY causes mostly beneficial effects in contrast to mammals in general and humans in particular.  

Reviewer 3 Report

The authors have done an excellent job in evaluating the utility of data capture from the medical record system which is transmitted to a third party for data abstraction.  The readers would be definitely interested to know the utility of the above-mentioned data collection method. 

Minor comments:

  • Please italicize all gene names
  • The term mutation is not preferred anymore. I would recommend switching the term mutation with “variant”

Major comments

  • 2.2: How many of these patients had a severe phenotype? Have they been on G tube feeding? Was anyone on ketogenic diet? All these factors could contribute to the fall in weight percentile as the disease progresses.
  • Microcephaly is a common finding with SLC13A5 related epileptic encephalopathy. It is interesting that none of your patients had microcephaly. Moreover, there are a few outliers in the growth parameters for a couple of the patients. Are the growth parameters obtained consistently from a single provider like their primary care provider? Or do they include measurements from different provider’s offices? (example: M2 : height 0-36 months)
  • Table 2 does not include any information re: their teeth anomalies including amelogenesis imperfecta which is a major part of the phenotype. Please include that information in table 2.

Round 2

Reviewer 2 Report

I am satisfied with the changes made by the authors. I would recommend the manuscript to be published.